# Automatic Recognition and Quantification Feeding Behaviors of Nursery Pigs Using Improved YOLOV5 and Feeding Functional Area Proposals

**DOI:** 10.3390/ani14040569

**Published:** 2024-02-08

**Authors:** Yizhi Luo, Jinjin Xia, Huazhong Lu, Haowen Luo, Enli Lv, Zhixiong Zeng, Bin Li, Fanming Meng, Aqing Yang

**Affiliations:** 1Institute of Facility Agriculture, Guangdong Academy of Agricultural Sciences, Guangzhou 510640, China; luoyizhi@gdaas.cn (Y.L.); wenyluyo@21cn.com (H.L.); 2State Key Laboratory of Swine and Poultry Breeding Industry, Guangzhou 510645, China; huazlu@scau.edu.cn (H.L.); enlilv@scau.edu.cn (E.L.); ganli180@sina.com (B.L.); mengfanming@gdaas.cn (F.M.); 3College of Engineering, South China Agricultural University, Guangzhou 510642, China; 2011010005@gdmec.edu.cn (J.X.); zhixzeng@scau.edu.cn (Z.Z.); 4Institute of Animal Science, Guangdong Academy of Agricultural Sciences, Guangzhou 510645, China; 5College of Computer Science, Guangdong Polytechnic Normal University, Guangzhou 510665, China

**Keywords:** nursery pigs, feeding behavior recognition, functional area proposals, behavioral quantification, transformer

## Abstract

**Simple Summary:**

In commercial-intensive pig farms, the identification and quantification of piglet feeding behavior is an important indicator for daily inspections. However, the model often judges pigs resting in the feeding area as the feeding behavior. Considering the functional characteristics of the pig house feeding area, we present a novel method based on the improved YOLOV5 and feeding functional area suggestions to identify the feeding and non-feeding behaviors of nursery piglets. In addition, four advanced models and ablation experiments are adopted for a comparative evaluation of the robustness of the model. This article proposes a model that can help the management team adjust piglet feeding patterns and save more manpower and feed.

**Abstract:**

A novel method is proposed based on the improved YOLOV5 and feeding functional area proposals to identify the feeding behaviors of nursery piglets in a complex light and different posture environment. The method consists of three steps: first, the corner coordinates of the feeding functional area were set up by using the shape characteristics of the trough proposals and the ratio of the corner point to the image width and height to separate the irregular feeding area; second, a transformer module model was introduced based on YOLOV5 for highly accurate head detection; and third, the feeding behavior was recognized and counted by calculating the proportion of the head in the located feeding area. The pig head dataset was constructed, including 5040 training sets with 54,670 piglet head boxes, and 1200 test sets, and 25,330 piglet head boxes. The improved model achieves a 5.8% increase in the mAP and a 4.7% increase in the F1 score compared with the YOLOV5s model. The model is also applied to analyze the feeding pattern of group-housed nursery pigs in 24 h continuous monitoring and finds that nursing pigs have different feeding rhythms for the day and night, with peak feeding periods at 7:00–9:00 and 15:00–17:00 and decreased feeding periods at 12:00–14:00 and 0:00–6:00. The model provides a solution for identifying and quantifying pig feeding behaviors and offers a data basis for adjusting the farm feeding scheme.

## 1. Introduction

Automating the recognition and quantification of feeding behaviors in group-housed pigs can greatly assist in the identification of potential health and welfare issues [1,2,3]. A divergence in the food intake of pigs has been frequently linked to disease existence [4,5,6]. The continuous monitoring of pig feeding behavior will aid in the design of an appropriate feed delivery plan to help breeders regulate production plans, reduce the waste of fodder, and embrace the production potential of pigs [7,8,9,10,11,12,13]. However, individual pigs foraging or resting in the feeding area are often judged by the model as NFB (no feeding behavior) without consuming any feed [14]. The key to solving this problem is understanding how to distinguish the behavioral differences between feeding behavior and non-feeding behavior. Considering the functional characteristics of the pig house feeding area, the accurate identification of pig heads in the feeding area is a prerequisite for the identification of feeding behavior [15,16].

Several researchers have previously investigated systems for monitoring pig feeding behavior [17,18,19,20]. The methods are mainly based on wearable sensors (e.g., accelerometers, magnetometers, etc.) and non-contact video monitoring. As for contact electronic sensors, for instance, an electronic feeding system based on radio frequency identification (RFID) technology can monitor the meal interval per day of sows to assess their health status by assessing an individual pig’s frequency of visits to feeders and water sources [13,21].  Kapun et al. proposed a feeding behavior recognition system for fattening pigs based on ultrahigh frequency (UHF-RFID) technology, which automatically reads the corresponding identity information when pigs are close to the feeding area or drinking area. The sensitivity of the system for feeding behavior and drinking behavior is 80% and 60%, respectively [13]. The above method used contact sensing radio frequency technology to identify the feeding and drinking behavior of pigs, but it is limited by the large labor cost required to install systems and tags and it remains somewhat invasive for pigs. Video monitoring is a low-cost, easy-to-maintain, and non-destructive monitoring technique. Zhu et al. identified individual pigs by calculating the Euclid distance between the pig and the standard sample and used it to determine whether the pig had drinking behavior based on the contact time between the pig and the drinking teat [22]. This method utilizes artificially designed features and is not accurate in other complex scenes. With the improvements in computing power, behavior recognition based on deep learning technology is widely used in the field of agriculture.

Compared with artificial features, the model based on deep learning can automatically learn the target features and improve the performance of the behavior recognition system [23,24,25,26]. A deep learning method with fully convolutional image processing was introduced to improve accuracy, which extracts the roundness of the head and the occlusion area between the head and the feeding area as spatial features based on the fully convolutional network (FCN), and it uses an optical flow vector to define the motion intensity of the head as a temporal feature, further improving the detection accuracy of the model [17]. However, current models such as Faster RCNN, YOLO, and CenterNet use a large number of proposals, anchors, or window centers, which are often less accurate in dense scenes. The transformer utilizes a self-attention mechanism for prediction, which performs well in the task of dense scene object detection. In addition, 3D cameras and thermal imaging cameras are also used in the field of animal behavior recognition [27,28,29,30,31,32,33].

In this work, we develop a single-view video analysis method to identify individual pigs and the numbers that engage in feeding behavior in large-scale pig herds without the need for additional sensors or individual markers. This approach differs from previous research in that it combines the location characteristics of the feeding area and the location of the pig head area where the feeding behavior occurs. In addition, a transformer module model was introduced based on YOLOV5 for pig head detection.

## 2. Materials and Methods

### 2.1. Animals, Housing, and Management

The experimental data in this paper were obtained from large-scale pig farms in Yunnan Province, China, from August to November 2021. The nursery barn of the pig farm consists of 28 pens with left and right symmetry (17 on one side), each 3.5 m long and 2.2 m wide see Figure 1a below. The front doors and partitions of the pens are made of PVC sheets, the floor of each pen is all plastic, and it is equipped with one pen of five feeders and two drinking fountains see Figure 1c–e below. Each unit is equipped with an environmental control device and ventilation system to regulate the indoor temperature and humidity see Figure 1f–j below. The lighting time in the house is generally from 8:00 to 12:00 in the morning and from 14:00 to 18:00 in the afternoon.

In addition, there are a total of 2404 nursery pigs in the nursery barn, and the piglets in each nursery barn are all crossbred pigs and the starting weight is about 6.2 kg. The average activity area of each piglet is 0.35 square meters, in which they can obtain sufficient water and feed. The feeding method of the traditional mechanical trough is to put more than the theoretically required feeding amount into the trough according to the age of the pig and put it twice a day at 8:00 and 15:00. The indoor air data logging method is to record the room temperature, and the average relative humidity is set to every 5 min through the PR-3003WS-X USB temperature (PRSENS, Jinan, China) and humidity integrated data recorder produced by the PRSENS merchant (PRSENS, Jinan, China). The average room temperature is about 26.6 °C, and the average relative humidity is about 67.7%. The RGB camera (BA12-H, BA, Guangzhou, China) is used to collect data, which is connected to the server through a wireless network video recorder (WNVR), and the video is recorded and stored on the hard disk at the same time see Figure 1b below.

### 2.2. Dataset

#### 2.2.1. Definition of Behaviors

Generally, the pig pen can be divided into five areas. As shown in Figure 2a, the red box ➀ is the drinking area, the yellow box ➁ is the feeding area, the orange box ➂ is the resting area, and the purple box ➃ and white box ➄ are the areas for excretion and being active, respectively.

Based on data from the animal ethology literature [34], the definitions of feeding behavior and non-feeding behavior are shown in Table 1, and examples of the two behaviors are shown in Figure 2b,c. In particular, compared to the feeding behavior, the spatial position of the NFB is disturbing. As shown in Figure 2b, the head of the pig (No. 7) does not enter the trough, but near the trough the head behavior is considered NFB.

#### 2.2.2. Data Acquisition and Preprocessing

Considering the application scenario characteristics of the model, 356 videos with differences in the skin color, light intensity, and posture of piglets were screened from all the data of a nursery barn. After video cutting pictures and data amplification, the final piglet head dataset is expanded to 6240, and the piglet head frame is about 80 K, including 5040 training sets and verification sets, with 54,670 piglet head boxes, 1200 test sets, and 25,330 piglet head boxes (Figure 3). The final data processing link is to use the interactive labeling tool Labelme (https://github.com/xwwwb/lableme_demo, accessed on 1 October 2021) to label the piglet head image, refer to the COCO dataset type, and export the .json file.

### 2.3. The Overall Process of Behavior Recognition and Quantification

According to the behavior description of the pig feeding process in Table 1, during the feeding process the head of a piglet will enter the trough, and the body will remain stationary (without interference from other piglets) until the end of the feeding process. Therefore, in combination with the division of the functional areas inside the pigsty, a novel method is introduced based on deep learning and functional area proposals; the overall process is seen in Figure 4.

#### Feeding Functional Area Proposals Filtering Strategy

A traditional closed-loop area separation method is adopted in this paper to extract the feeding functional area proposals in the image (Algorithm 1). First, four coordinate points, L1, L2, R1, and R2, are selected according to the shape characteristics of the feed trough to represent the feeding area in the image. The location of the feeding area is shown in Figure 5a. Secondly, the closed feeding area A and the non-feeding area B are separated by setting the ratio of the coordinate points to the width and height of the image, as shown in Figure 5b. Finally, behavioral identification and quantity statistics are performed based on whether the piglet’s head is in the feeding area and the number of piglets.
**Algorithm 1** Feeding functional area proposals filtering strategyinput: Pig head detection bounding box B[1…n]output: The number of feeding and drinking in the current frame $Num_{feeding}$1:$Num_{feeding}\leftarrow 0;n\leftarrow 1$2:**for** i = 1 : Len(B[n]) **do**3:      **if** $B_{1}\;Area\left[Feeding\right]$ **then**4:          $Num_{feeding}\leftarrow Num_{feeding}\;+\;1$5:      **end if**6:**end for**7:**return** 
$Num_{feeding}$
where B[1…n] means all the bounding boxes are in the pig pen area, $Num_{feeding}$ means the number of heads with feeding behavior, and Area[Feeding] means the extent of the feeding area.


**Figure 5 animals-14-00569-f005:**
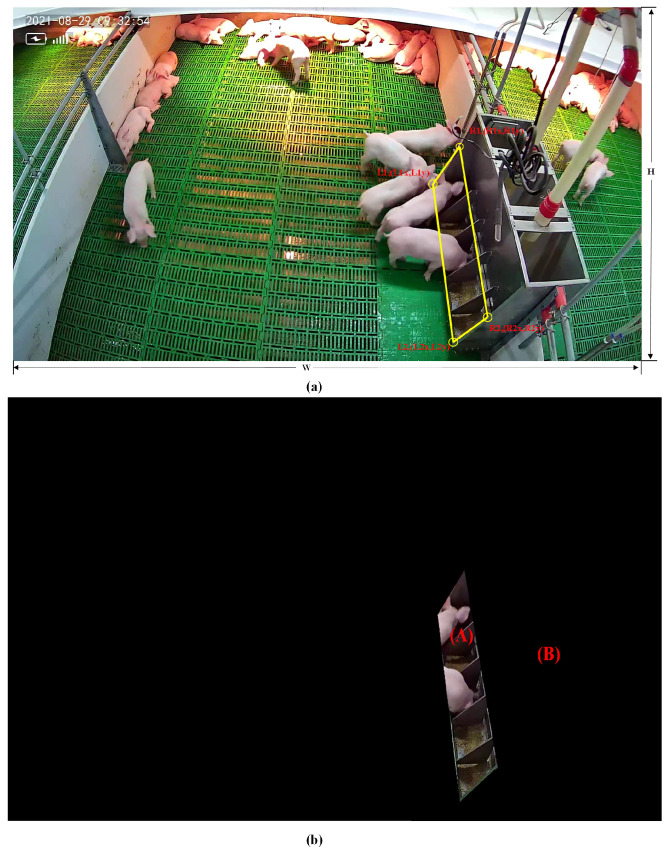
Example of feeding area location: (**a**) trough corner selection and (**b**) region of interest that may be feeding area.

### 2.4. Piglet Head Detection Network Architecture

Aiming to address the problems of piglet head occlusion, complex light in detecting feeding behavior, this paper proposes an improved deep learning model based on tph-YOLOV5 to realize piglet head detection in the region of interest for further characterizing the feeding behavior of piglets. Compared with the YOLOV4 model, YOLOV5 adopts the method of sharing convolution kernels to reduce the number of calculations, and reduce the number of model parameters, and improve the efficiency of network computing. The improved model architecture is shown in Figure 6, which is mainly composed of three parts, including the backbone feature extraction network, the neck network, and the prediction head.

This study follows the main framework of tph-YOLOV5, as shown in Figure 6. First, the main structure adopts the C3 block (light gray block), which iterates 3, 9, and 9, respectively, as shown in Figure 6 ➀, ➁, and ➂. Subsequently, a transformer module is added at the end of the backbone network to improve the ability to capture the features of the pig’s head, as shown in Figure 6 ➃ (light green block). On the one hand, this module enhances the ability of local information features of video frames [35]; on the other hand, it makes full use of context information to improve the feature extraction ability of the piglets’ heads.

In the feature fusion part, a path aggregation network [36] is used to optimize feature reuse. Specially, upsampling is performed three times, respectively (Figure 6 ➄, ➅, and ➆), and the convolutional block attention module (CBAM) is integrated into YOLOv5, which helps the model to find areas of interest in large-scale images (yellow block, Figure 6 ➆, ➇, and ➈).

For the prediction head, different scales of feature maps are obtained by the operation of four 1 × 1 convolutions, as shown in Figure 6c. The pig is a non-rigid animal, and its head rotation, occlusion, distance from the camera, and other factors will affect the size of the piglets in the image. Therefore, the transformer module is added at the end of the prediction header. Moreover, the number of prediction headers is increased from three to four.

#### YOLO Principle

In practice, there are two types of target detection models: the two-stage model and one-stage model. Among them, Faster RCNN is a two-stage model that is used more in current research. The model introduces a region generation network (Region Proposal Network, RPN). Compared with RCNN, this model extracts the features of the input image and generates candidate regions at the same time to avoid duplication. Different from the Faster RCNN model, the YOLO series treats object detection as a regression problem and directly derives the piglet head position, category, and confidence level of the piglet head detection [37]. The structure is shown in Figure 7.

Initially, an image is divided into S × S grids, K bounding boxes are generated for each grid-based anchor box, and C represents the posture category of the pig; each bounding box contains six predicted values: x, y, w, h, confidence, and category probability (Pr). Among them, x, y, w, and h are used to describe the position of the bounding boxes, where the confidence represents whether the predicted box contains the object and the accuracy of the prediction box, and the pig posture category probability map is simultaneously generated. As stated in Formula (1), the confidence is the product of Pr and IOU. If the ground truth falls into this grid cell, $Pr_{object}$ takes a value of 1; otherwise, it is 0. IOU is the intersection ratio between the bounding box and the actual ground truth, and the model employs non-maximum suppression to search for inhibiting redundant boxes.
(1)$$\begin{array}{c} Pr\left(Class_{i}|Objection_{i}\right)\times Pr\left(Object\right)\times IOU_{pred}^{truth}=Pr\left(Class_{i}\right)\times IOU_{pred}^{truth} \end{array}$$
where $Pr_{object}$ is ∈{0,1}.

The Intersection over Union (IOU) loss function is used for boundary box prediction to reflect the detection effect of the piglet head. As shown in Figure 8, the green box represents the real box and its area *a*, the blue box represents the prediction box and its area *b*, *I* is the intersection of the green box area and the blue box area, *U* is the union of the green box area and the blue box area, and $Loss_{iou}$ is the loss function, as shown in Formulas (2) and (3).
(2)$$\begin{array}{cc} IOU & =\frac{I}{U} \end{array}$$
(3)$$\begin{array}{cc} Loss_{iou} & =-\ln(IOU) \end{array}$$
where $\tilde{x}=\left(\tilde{x_{t}},\tilde{x_{b}},\tilde{x_{l}},\tilde{x_{r}}\right)$ represent the distances between the current pixel location (i, j) and the top, bottom, left, and right bounds of the ground truth, respectively. For simplicity, we omit footnote i, j in the rest of this paper. Accordingly, a predicted bounding box is defined as $x=(x_{t},x_{b},x_{l},x_{r})$.

The occlusion area, center point distance, and aspect ratio relationship between the predicted frame and the actual frame is taken into account in the CIOU loss function based on the DIOU; the formulas are shown in (4)–(6), and the schematic diagram of the CIOU computing structure is shown in Figure 9.
(4)$$\begin{array}{cc} Loss_{ciou} & =1-IOU+\frac{\rho^{2}\left(b,b^{gt}\right)}{C^{2}} \end{array}$$
(5)$$\begin{array}{cc} \alpha & =\frac{v}{(1-IOU)+v} \end{array}$$
(6)$$\begin{array}{cc} v & =\frac{4}{\pi^{2}}{\left(\arctan\frac{w^{gt}}{h^{gt}}-\arctan\frac{w}{h}\right)}^{2} \end{array}$$
where b and $b^{gt}$ represent the center points of the prediction box and the real box, respectively, and ρ represents the calculation of the Euclidean distance between the two center points, and C represents the diagonal distance of the minimum closure area that can contain both the prediction box and the true box, α is the parameter of the trade-off, and v is the parameter that measures the consistency of the aspect ratio.

### 2.5. Evaluation Index

The mean average precision (mAP) and the F1 score (F1_score) are adopted to evaluate the effect of the pig head detection model in this paper [38]. The formulas for the mAP and the F1_score are as follows:(7)$$\begin{array}{cc} Precision & =\frac{TP}{TP+FP} \end{array}$$(8)$$\begin{array}{cc} Recall & =\frac{TP}{TP+FN} \end{array}$$(9)$$\begin{array}{cc} AP & =\int_{0}^{1}Precision\times Recalldr \end{array}$$(10)$$\begin{array}{cc} mAP & =\frac{AP}{n} \end{array}$$(11)$$\begin{array}{cc} F1-score & =\frac{2(Precision\times Recall)}{Precision+Recall} \end{array}$$
where *n* represents the classes, n=1.

### 2.6. Training Environment and Equipment Description

In this paper, we adopt the Pytorch framework (https://pytorch.org/, accessed on 1 October 2021). The training environment and equipment description are shown in Table 2.

The initialization weights play an important roll in the process of network training. The hyperparameter optimization method is used in YOLOV5 [39], which uses a genetic algorithm for better model performance [40]. In this study, we use pretrained imageNet hyperparameters to fine-tune the pig head dataset.

## 3. Results

### 3.1. Performance Comparison of Different Models

Video analysis is one of the main daily tasks of managers. A novel feeding behavior recognition and quantification model was proposed in this paper to help administrators extract effective information from a large scale of similar videos for the timely detection of abnormalities in pig herds. Simultaneously, four advanced models were adopted for a comparative evaluation on the pig head dataset, including YOLOV3 (DarkNet53), YOLOV5 (CSPDarknet53), YOLOV5 (MobileNetV3-Large), and YOLOV5 (RepVGG), as detailed in Table 3. Compared with the yolov5 model, the model in this article improved the mAP index and F1_score index by 5.8% and 4.7%, respectively.

In addition, we analyzed the confidence loss (Figure 10a) and bounding box loss (Figure 10b) after 300 epochs. The loss function is one of the main indicators for model performance evaluation. It can be seen that with the increase in the number of training epochs, the loss values of both models show a steady downward trend, and the curves are smoother. The reason for the rapid decline in the category loss was attributed to a single category (piglet head).

In particular, the two models were basically stable in the first 120 epochs, and the loss of the proposed model dropped faster and lower. In addition, the mAP0.5, the precision rate, and the recall rate were evaluated, and they are reported in Figure 11. The results show that the evaluation indicators of the proposed model are better than for YOLOV5.

### 3.2. Ablation Test

In order to evaluate the effectiveness of each improvement to the model proposed in this paper, ablation experiments were conducted to analyze the causal relationship of each improved component. The experiment was divided into five groups, namely, M0–M4. The test results are shown in Table 4.

### 3.3. Performance Comparison of Three IOU Loss Functions

Moreover, three IOU loss functions were used for the comparative analysis in this paper, including Generalized IOU (GIOU loss), Distance IOU loss (DIOU), and Complete IOU loss (CIOU). The results are shown in Table 5. The CIOU loss function performed best in the model performance matrix, with the mAP and F1 being 0.921 and 0.90, respectively. In the regression of the boundary box, the coupling characteristics between the size of the piglet head boundary and its position were better coordinated.

## 4. Discussions

### 4.1. The Impact of Different Models on the Pig Head Detection Performance

A representative sample is selected in this work to compare the performances of different models in recognizing pig heads. Figure 12b (orange elliptical area, left) shows a complex lighting scene (heat preservation lamp). An interesting phenomenon is that no false detections occurred under the occlusion of the pig’s head and forelimbs (black area in Figure 12b). This also reflects the powerful ability of deep learning in feature extraction. Figure 12c (purple elliptical area, right) shows the situation of the occlusion scene. The comparison results show that some lightweight models (e.g., MobileNetV3-Large) perform poorly in obstructed mountains, and their ability to extract features from pig heads is not strong. In addition, target occlusion of the same category is a specific challenge in agricultural target detection applications. Missed detection will occur due to the non-maximum suppression (NMS) operation to remove redundant frames, also representing a future research direction.

### 4.2. Results of Feeding Behaviors and No Feeding Behaviors

We randomly selected 24 continuous hours of video data and saved the test results in a .txt file containing the identification results of the feeding behavior and no feeding behavior and their corresponding quantities.

Figure 13 shows examples of head detection in FB and NFB recognition and the corresponding heatmap [41]. The Grad CAM method was introduced to use the gradient of the object class relative to the feature map to visualize the class differentiation and location in the input image. As shown in Figure 13b,d, the red areas in the heatmap represent the regions of interest in the model. The pigs’ heads were activated in the heatmap, reflecting the pig head detection ability of the model.

A previous study extracted spatiotemporal features from videos to recognize the feed behaviors of nursery pigs and further improved the recognition accuracy by changing the video length and frame rate [19]. In contrast to the task of video classification, this paper focused on the detection of pig feeding behavior at the image frame level, which can be used to quantify the herd behavior of pigs and reflect their health status.

As shown in Figure 13c, the pig is considered as having NFB even though there are piglets near the trough. In addition, compared to the detection of the heating lamp area (complex lighting conditions), the result performance is better in the feeding area. Lighting conditions have always been a huge challenge for image processing. One study used a 3D camera to handle sow pose detection under complex lighting conditions, with significant advantages in resisting lighting and spatial coordinate relationships. Compared with RGB cameras, 3D cameras have smaller viewing angles and are more expensive.

#### Distribution of Feeding Time of Piglets in Different Periods for All Weather

Figure 14 shows the distribution of the feeding and drinking time of the piglets across different all-weather periods. From the total consumption trends of feeding and drinking, the piglets ate and drank at the same time. The total consumption time of food and water for the piglets demonstrated a similar change trend, including the three feeding peaks (5:00–6:00 GMT, 15:00–17:00 GMT, and 20:00–21:00 GMT). The feeding behavior of the pigs increased in the period from 7:00 to 8:00 GMT, when most of the piglets woke up. From 10:00 to 11:00 GMT, their feeding behavior decreased slightly. Before the noon break, their feeding behavior increased. After 12:00 GMT, the keepers turned off the fluorescent lamp, and the piglets began to rest. Their feeding behavior gradually decreased. From 14:00 to 15:00 GMT, a large number of piglets woke up to feed, and, until 18:00 GMT, the fluorescent lamp was turned off. The last peak appeared at night when the keepers were patrolling. Therefore, the feeding behavior of the piglets demonstrated clear day and night characteristics, with intermittent feeding, and was affected by the feeding environment.

In addition, there was a statistical distribution of the maximum number of piglets in the all-weather feeding periods. As shown in Figure 15, the peak and trough of the feeding appeared alternately, indicating that the food intake of the pigs was phased. The red line represents the feeder space. It can be seen from the trend of the curve in the figure that the number of piglets feeding was larger than that of the feeder space in the periods from 5:00 to 12:00 GMT, 15:00 to 16:00 GMT, and 20:00 to 21:00 GMT, and it exceeded it by 50% in the periods from 7:00 to 9:00 GMT and 20:00 to 21:00 GMT. In addition, the number of piglets feeding was lower than the feeder space at 22:00–5:00 GMT. During this period, only a few piglets fed intermittently, as the feeding peak often occurred before and after the pigs slept; this provides a reference for breeders when developing feeding plans.

## 5. Conclusions

Continuous automated monitoring of pig feeding behavior can help breeders standardize production plans, reduce feed waste, and give full play to pig production potential. However, manual inspection statistics are a heavy workload and highly subjective, which is not conducive to production needs. This work proposes a new method based on the improved yolov5 and feeding functional area suggestions to identify feeding and non-feeding behaviors of nursery piglets, simplifying complex feeding behaviors into a functional area piglet head detection problem. Particularly, in piglet head detection, compared with YOLOV5, the improved model achieved a mAP increase of 5.8% and an F1_score increase of 4.7%. At the same time, the model was applied to the analysis of the feeding patterns of nursery pigs raised in groups for 24 h of continuous monitoring. It was found that the feeding rhythms of nursery pigs are different in the day and night. The peak feeding period is 7:00–9:00 and 15:00–17:00. And the feeding time is from 12:00 to 14:00 and 0:00 to 6:00. This model provides a solution for identifying and quantifying pig feeding behavior and provides a solution for adjusting pig farm feeding programs that provide the basis for the data. In addition, according to the effect of model detection, objects of the same category will be missed in detection due to the NMS operation to remove redundant frames, which is also a future research direction.

## Figures and Tables

**Figure 1 animals-14-00569-f001:**
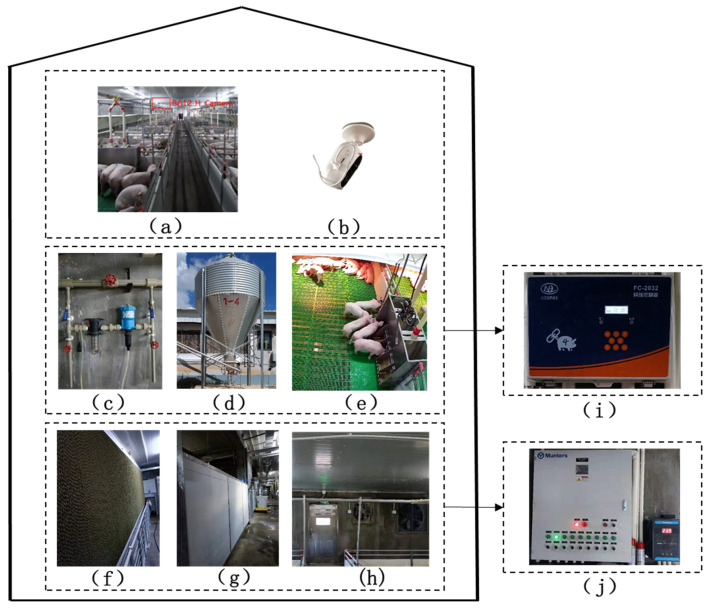
Description of the barn. (**a**) Inside the barn. (**b**) Camera placement area. (**c**) Water and drug mixing pipeline. (**d**) Feed supply. (**e**) Top view of pig pen. (**f**) Cooling wet curtain. (**g**) Air baffle. (**h**) Variable speed fan. (**i**) Environmental control controller. (**j**) Equipment master controller.

**Figure 2 animals-14-00569-f002:**
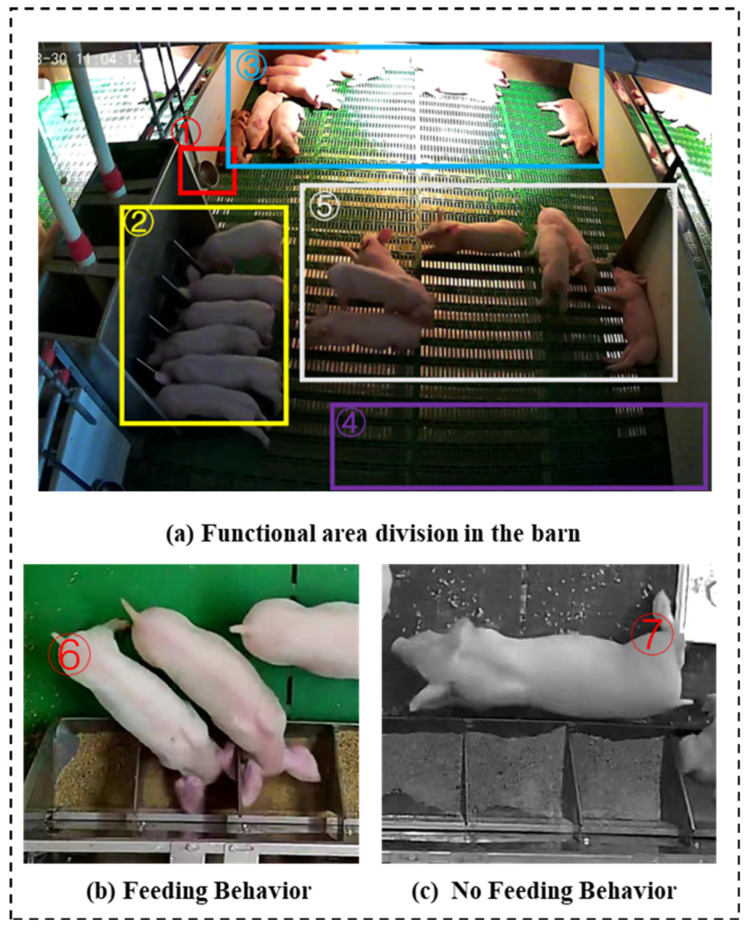
Schematic diagram of functional area distribution and behavior. (**a**) ➀ The drinking area, ➁ the feeding area, ➂ the resting area, ➃ the excretory area, and ➄ the active area.

**Figure 3 animals-14-00569-f003:**
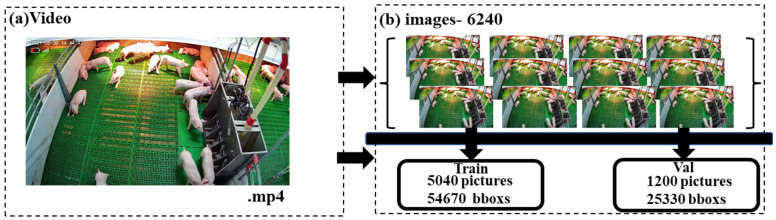
The constitution of our dataset.

**Figure 4 animals-14-00569-f004:**
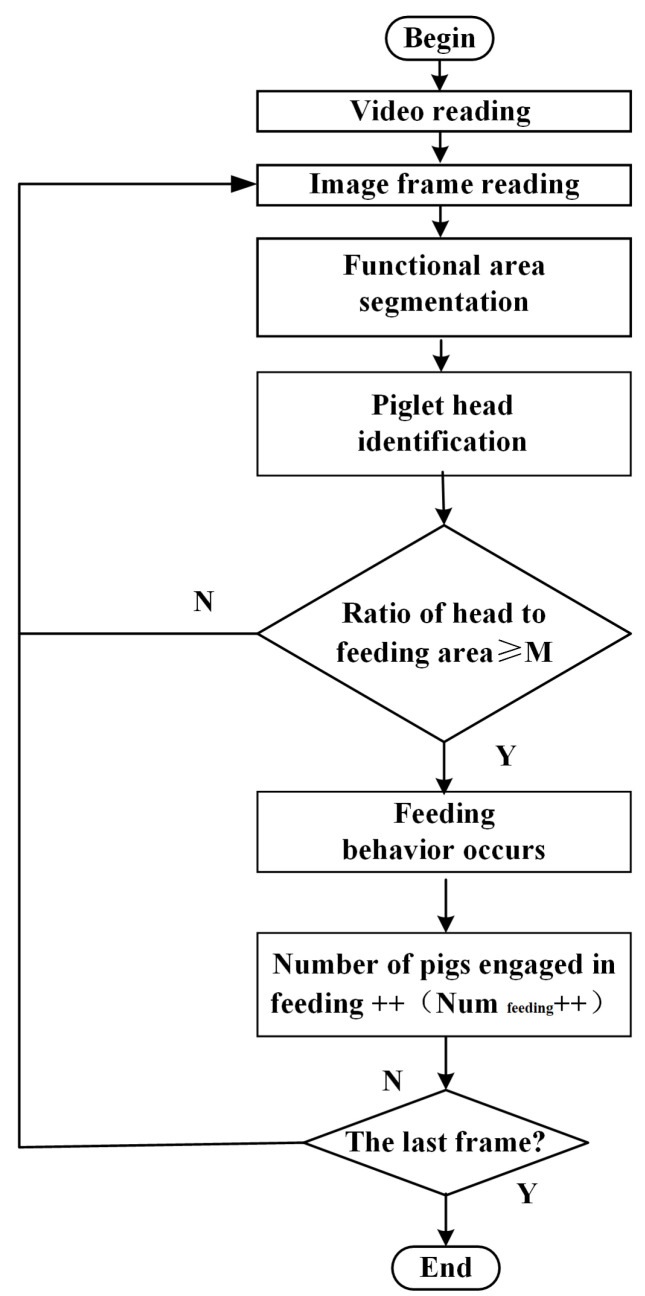
The overall process.

**Figure 6 animals-14-00569-f006:**
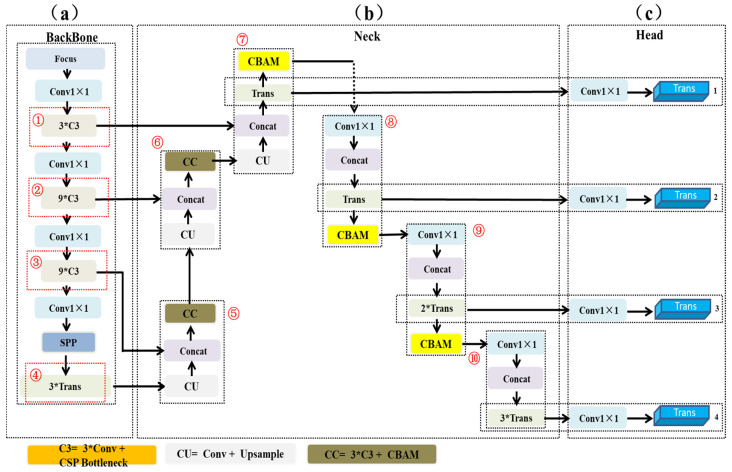
Piglet head detection network architecture: (**a**) the backbone of the model (**b**) the neck of the model and (**c**) the head of the model.

**Figure 7 animals-14-00569-f007:**
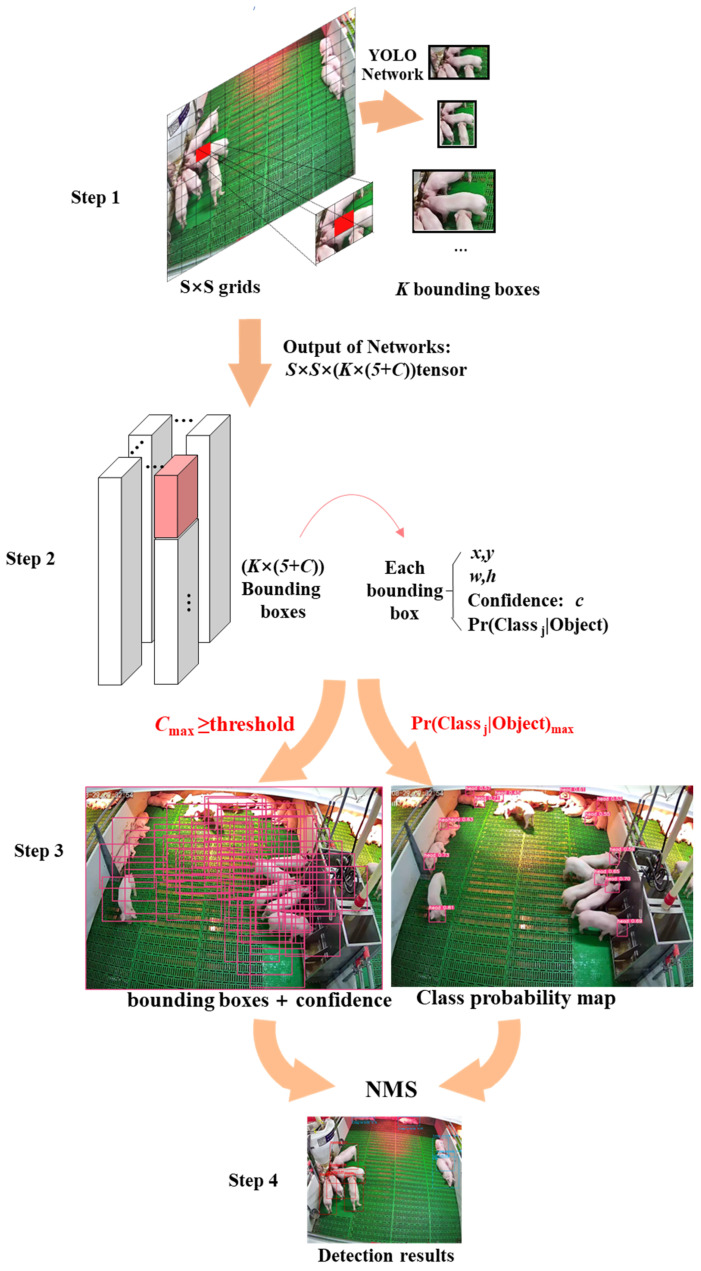
The primary principle of YOLO.

**Figure 8 animals-14-00569-f008:**
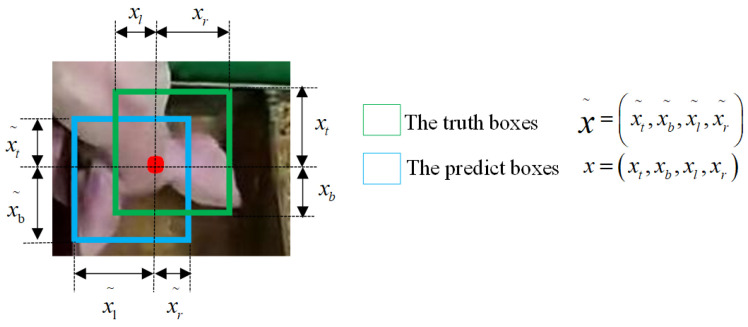
Schematic diagram of IOU prediction box loss.

**Figure 9 animals-14-00569-f009:**
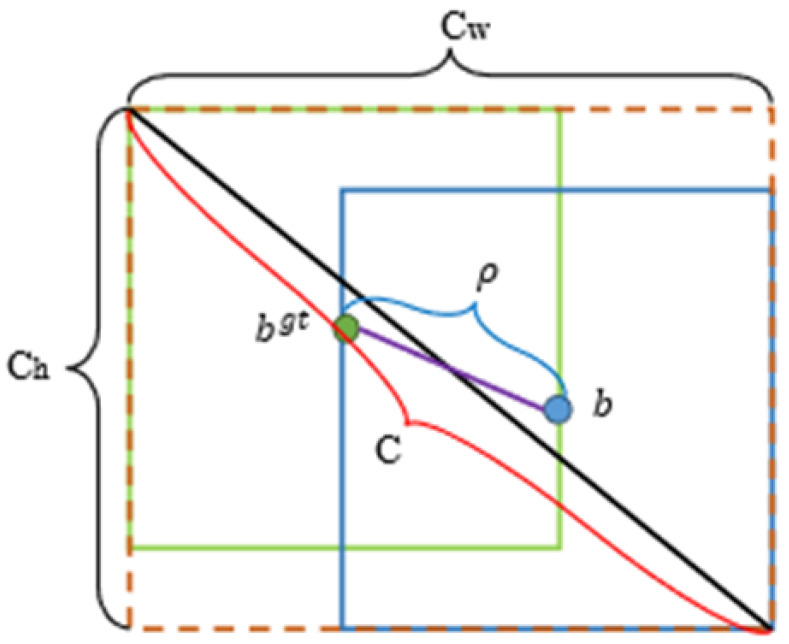
Schematic diagram of CIOU computing structure.

**Figure 10 animals-14-00569-f010:**
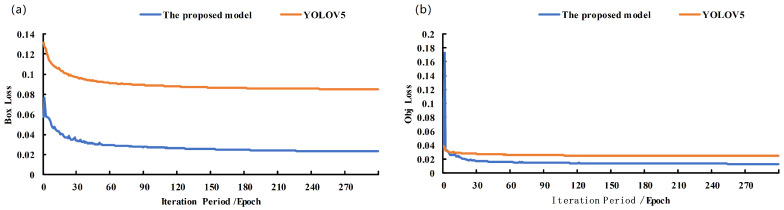
Train loss for the YOLOV5+Transform (the proposed model) and YOLOV5. (**a**) Box loss. (**b**) Obj loss.

**Figure 11 animals-14-00569-f011:**
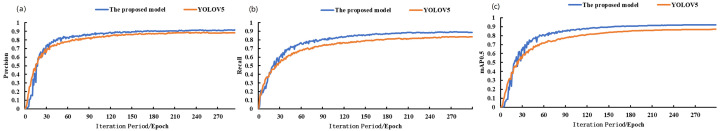
Train loss for the proposed model and YOLOV5. (**a**) Precision. (**b**) Recall. (**c**) mAP0.5.

**Figure 12 animals-14-00569-f012:**
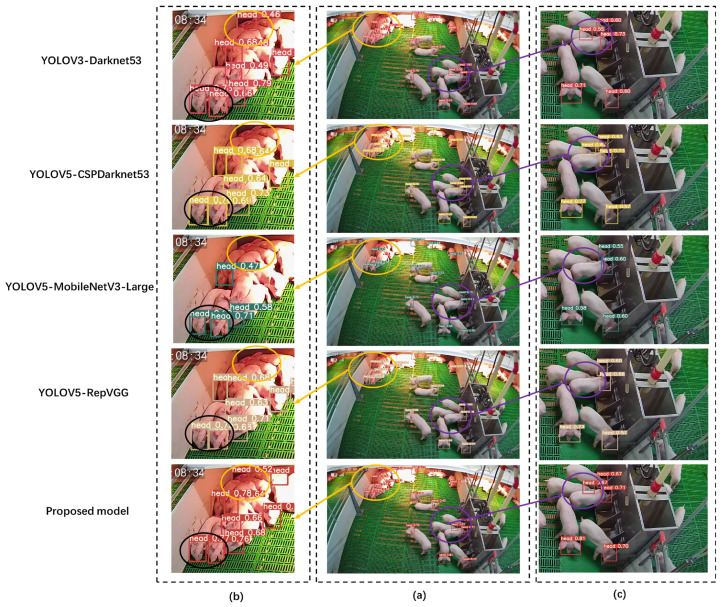
Detection effects of different models. (**a**) Example. (**b**) Partial enlargement 1 of the example. (**c**) Partial enlargement 2 of the example.

**Figure 13 animals-14-00569-f013:**
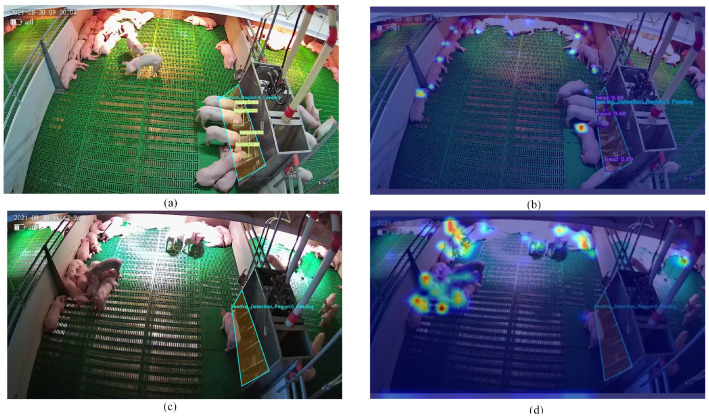
Behavior quantification examples and category heatmap visualization. (**a**) Feeding behavior. (**b**) Heatmap visualization of feeding behavior. (**c**) Not feeding behavior. (**d**) Heatmap visualization of no feeding behavior.

**Figure 14 animals-14-00569-f014:**
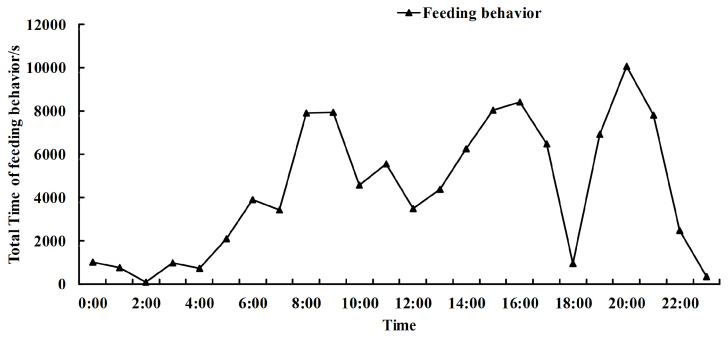
Feeding time of all piglets in different periods (all weather).

**Figure 15 animals-14-00569-f015:**
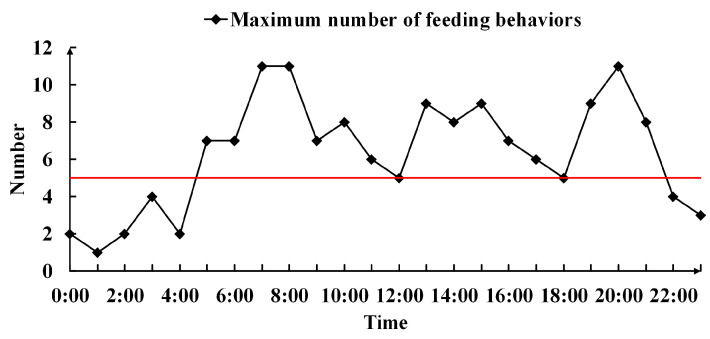
Maximum number of piglets feeding behavior (all-weather).

**Table 1 animals-14-00569-t001:** Definition of pig behaviors.

Category	Abbreviation	Description
Feeding Behavior	FB	The piglet’s head enters the trough, the head droops, and the two forelimbs enter or are near the trough.
Not Feeding Behavior	NFB	The piglet does not enter the trough, its head is not in the trough, its body is close to or away from the trough, and it stands on the floor vertically on all four limbs or lies on its side to rest.

**Table 2 animals-14-00569-t002:** Training environment and equipment description.

Configuration	Parameter
Image resolution	2304 pixels ∗ 1296 pixels (W ∗ H)
Training framework	Python 3.7 programming language, Pytorch framework
Pretrained model	ImageNet model
Operating system	Ubuntu18.04 version
Accelerated environment	CUDA11 and CUDNN 7
Development environment	Vscode
Computer configuration used in training and testing	Dell Tower Workstation Intel@Xeon(R) T7920 Processor, 32 GB RDIMM, 512 G Solid State Drive, 4 TB Mechanical Hard Drive, Graphics Card RTX2080Ti

**Table 3 animals-14-00569-t003:** Performance comparison of different models.

Detector	Backbone	mAP	F1_Score
YoloV3	Darknet53	83.1%	82.0%
YoloV5	CSPDarknet53	87.0%	86.0%
YoloV5	MobileNetV3-Large	66.4%	66.0%
YoloV5	RepVGG	78.0%	77.0%
YoloV5	proposed model	$\begin{array}{c} 92.1\% \end{array}$	$\begin{array}{c} 90\% \end{array}$

**Table 4 animals-14-00569-t004:** Ablation test.

Classes	C3TR	CBAM	Change Head	mAP	F1_Score
M0	-	-	-	87.0%	86.0%
M1	✓	-	-	89.4%	87.0%
M2	-	✓	-	87.2%	86.0%
M3	-	-	✓	87.4%	85.0%
M4	✓	✓	✓	92.1%	90%

M0 = YOLOv5s, M1 = YOLOv5s + C3TR, M2 = YOLOv5s + CBAM, M3 = YOLOv5s + changed head, and M4 = YOLOv5s + C3TR + CBAM + changed head.

**Table 5 animals-14-00569-t005:** Model performance results for different IOUs.

Model	IOU	mAP	F1_Score
The proposed model	GIOU	87.2%	87%
The proposed model	DIOU	89.4%%	87%
The proposed model	CIOU	92.1%%	90%

## Data Availability

Data are contained within the article.

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
