# Peer review of "Automatic Recognition and Quantification Feeding Behaviors of Nursery Pigs Using Improved YOLOV5 and Feeding Functional Area Proposals"

_animals, 2024, doi:10.3390/ani14040569_

Round 1
Reviewer 1 Report
Comments and Suggestions for Authors
Manuscript ID: animals-2788776
Title: Automatic recognition and quantification feeding behaviors of nursery pigs using improved YOLOV5 and feeding functional area proposals
This manuscript proposed a novel method based on YoloV5 to identify the feeding behaviors of nursery piglets in a complex light and different posture environment. As a result, this improved model achieves a 5.8% increase in mAP and a 4.7% increase in F1 score compared with the YOLOV5s model. This result could provides a automatic way for identifying and quantifying pig feeding behaviors.
Figure 1a. Please translate Chinese into English.
If possible, it is better to use this novel model to other species,e.g., cattle.
As you documented in Acknowledgments, it is better to deposit your code and raw data in a public database, e.g., githu, such that other researchers can benefit from your research.
Author Response
Cover Letter
Dear Editor :
Thanks for the great amount of time spent processing my manuscript, which we have carefully revised based on the editor's comments.
We would like to submit the enclosed manuscript entitled “Automatic recognition and quantification feeding behaviors of nursery pigs using improved YOLOV5 and feeding functional area proposals”, which we wish to be considered for publication in “animals”. No conflict of interest exists in the submission of this manuscript, and the manuscript is approved by all authors for publication. I would like to declare on behalf of my co-authors that the work described was original research that has not been published previously, and is not under consideration for publication elsewhere, in whole or in part. All the authors listed have approved the manuscript that is enclosed.
The major highlights in this paper can be stated as follows:
1) A novel method is proposed based on improved yolov5 and feeding functional area proposals to identify the feeding behaviors of nursery piglets in a complex light and different posture environment.
2) the corner coordinates of the feeding functional area were set up by using the shape characteristics of the trough proposals, and the ratio of the corner point to the image width and height to separate the irregular feeding area.
3)A transformer block model was introduced based on yolov5 for a accurate head detection.
I hope this paper is suitable for special Issues “Mathematical Modeling and Computer Vision in Animal Activity or Behavior”.
We deeply appreciate your consideration of our manuscript, and we look forward to receiving comments from the reviewers. If you have any queries, please don’t hesitate to contact me at the address below.
Thank you and best regards.
Yours sincerely,
Yizhi Luo
E-mail: luoyizhi@gdaas.cn
Address: Institute of Facility Agriculture, Guangdong Academy of Agricultural Sciences, Guangzhou 510642, China

Reviewer 2 Report
Comments and Suggestions for Authors
This research article proposed a novel method based on improved Yolo V5 with the feeding functional area proposals to identify the feeding behaviors of nursery piglets in a complex light and different posture environment. This paper is well organized, and the descriptions have flown well. To improve the quality of the paper, the authors need to consider the following items for revision:.
Major comments:
1. It is suggested that the author revises English writing.
2. Please check author guidelines for formatting and revision based on journal requirement. (Figure, table, line, spacing, etc).
3. Introduction is well flown, and the objectives are clear. Moreover, author may make a more clear statement about the target based on previous limitations as this is very common in the field of deep learning work.
4. The methods is well explained. It is needed to explain why author choose YOLO V5? Also need to be more details about the proposed structure part of the Yolo V5 model.
5. The results are well flown and explained with figures and tables. Author compare the results with different version of Yolo Vs. What about the other method such as SSD. Also the robustness of the proposed method need to be discussed. The discussion need to be revised with major finding and limitations.
6. If the software or platform (Labelme Tool, etc.) are not developed by author, it is recommended to add references.
7. Conclusion can be revised with major findings, drawbacks and future study.
Minor comments:
1. In Line 8, “The pig head dat set was” will be The pig head data set was”. Also revise this types of error throughout the text.
2. In line 36, Ref.[13] proposed a feeding behavior recognition. check if the writing style match with the author guidelines and revise similar issues.
3. Figures (Figure 1,3, etc) can be enlarged to get more clear view.
Comments on the Quality of English LanguageMinor editing of English language required
Author Response

(The authors gave the same response as above.)

Reviewer 3 Report
Comments and Suggestions for Authors
In the entire article, I did not find a more detailed description of the concept of your study. Please provide more information about the goals of your experiment, where you clarify what its subject is. Are you focusing solely on detecting the heads of piglets during feeding, i.e., in relation to the feeder? Are you detecting only the number of fed piglets, or are you going to the level of individual piglets? The article often mentions the behavior of piglets. It would be good to clarify what you actually evaluated and what is just an evaluation of the results in relation to the stated conclusions about behavior. In the study description, information about the camera used is missing – is it a monocular/stereoscopic camera, does it have night recording capability, and how do you evaluate the temperature of the piglets? Please add information about the camera's placement. Can the model also be applied to adult individuals or other types of animals? What would need to be changed?
Specify what is the input for processing (likely frames extracted from video), and what is the output. Please also mention the system's weaknesses and what needs to be ensured for the system's application in a specific hall.
In the detection of feeding, are you based on standardized dimensions and shapes of the feeder, or with each new feeder, will the model need to be retrained?
Have you considered a method based on weighing the feed and average consumption per piglet?
In addition to this, I have the following comments:
In line 33 – specify what sensors are used.
In line 35 – does the RFID device on the pig bother them? When using RFID, is it about the precise localization of the pig or just approaching the RFID device to the feeder – in the sense of counting piglets? The use of RFID has issues with signal interference near metallic objects, which could be a significant problem in the case of breeding stations and feeders.
In line 36 - Explain how the health status of a pig can be evaluated using RFID.
In line 40 - Why does the system have lower sensitivity in drinking identification?
In lines 41 and 42 - The issue with both mentioned methods is metallic constructions.
In line 43 - Video monitoring – how do you deal with a larger number of pigs at the feeder when multiple pigs are present simultaneously, leading to significant pig overlap?
In line 64 - "a single-view video analysis" - using only one camera may exacerbate the issue of pig overlap. Why limit it to one camera? Is computational time a reason? It would be good to supplement the situation where you indicate the camera's location in the hall relative to the feeder + provide recommendations for camera placement.
In lines 64-65 - "to identify individual pigs" - can you identify each pig individually, meaning repeatedly finding a specific individual?
In line 69 - "a highly accurate head detection" - again, what is your concept, why do you need to know the precise position of the head? Isn't it enough to know that the head is in the feeder, defined by corner points?
In line 73 - In most images, there is a smaller number of piglets; have you tried applying the model to dataset with a larger number of piglets? It would be beneficial to include information about the placement and orientation of the cameras.
In line 75 - What about a different space? Could you generally describe a model situation, listing everything needed for the model to function and providing a time estimate for deploying the system?
In line 81 - "2,404 nursery pigs" - with 14 farrowing crates, it would be 171 pigs per crate. Please explain why there is a much smaller number of pigs visible in the images.
Figure 1 - "b) Camera placement area" - The image doesn't allow identification of where the camera is placed, which is crucial information. What type of camera is used? Is it a monocular or stereoscopic camera? What are the recording parameters?
Figure 2a - The orange frame in the image blends with the surroundings; try using a different color (e.g., cyan). What is the subject of detection? Apart from detecting piglet heads, are functional areas 1-5 also part of the detection on each frame?
In line 107 - "Considering the actual needs of the model," specify the model's needs and requirements. Do you need to identify specific piglets (repeatedly the same individual), or is it about recognizing pigs as objects in general?
In lines 107-108 - "differences in skin color, light intensity, and posture of piglets" - does this refer to the piglet's head?
In line 109 - "After video cutting pictures and data amplification" - I assume you export images and then work solely with the images?
In line 113 - "tool Labelme" - cite the source for the program. It would be appropriate to add information about how Labelme is used in the processing.
In lines 117-118 - "body will remain stationary (without interference from other piglets) until the end of the feeding process." - is this always the case?
Figure 4 - how are functional areas utilized in your model?
In lines 124-126 - "four coordinate points L1, L2, R1, and R2 were selected according to the shape characteristics of the feed trough to represent the feeding area in the image." - it would be appropriate to mark them in the image. If these points represent the corners of the feeding trough, and piglets are there, not all of them might be visible. What happens then? Are you assuming a fixed position of the camera and trough, enabling you to consistently locate these points? Please explain. Are you working in 3D, and how were these coordinates obtained? Are you using a stereo camera?
Algorithm 1 - variables in the pseudocode are not explained anywhere.
Figure 5 - Images a and b need to be enlarged; the text in the images is unreadable. The feed trough cutout on the right is not visible.
In lines 133-134 - "problems that piglet head occlusion, complex light, and facial pollution" - How do you address these issues, especially with head occlusion problems when multiple piglets are at the feeder?
Figure 6 - The texts in the image are too small. The image description should also include explanations/descriptions for parts a, b, c.
In line 142 - Why did you choose to use YOLOv5?
In line 158 - I did not find information about the camera's placement in the article, and I also did not ascertain whether the issue of positioning multiple piglets at the feeder and piglet overlap is addressed.
Figure 7 - Image in step 4 - purple boxes are not visible. The same applies to step 3 on the left - overlapping boxes are poorly readable.
In line 172 - "the posture category of the pig" - probably referring to the head position of the pig, not the entire pig's posture. What are the possibilities?
In line 173 - How do you make predictions? Do you use the Kalman filter (KF)?
In lines 173-174 - "contains six predicted values: x, y, w, h, confidence, and category probability (Pr)" - Variables are not explained. I assume these represent the center position and dimensions of the bounding box, but it should be explicitly stated. What is the difference between confidence and probability?
In line 185 - Is the real box the result of the model's detection? How do you obtain the prediction box?
Figure 8 - What does the red dot in the image represent, and how is it obtained? Does it refer to "center point distance"? The text accompanying the dimensions should be larger.
In lines 189-190 - Variables CIOU and DIOU - Equations 4-6 do not use them. Only the variable IOU is present.
Figure 9 - There seems to be a missing reference to this figure in the text, despite the figure being commented on the previous page. Include a reference to Figure 9 in the text, and in the figure, add the red dot from Figure 8 to clarify the process of its identification. Does it represent the intersection of the black and purple lines?
In line 197 - "(F1-score)" - As this is the name of a variable, it is better to present it either as F1 score or F1_score.
In equations 7 and 8 - Provide an explanation of how the counts of True Positives (TP), True Negatives (TN), False Positives (FP), and False Negatives (FN) were determined in equations 7 and 8.
In Table 2 - "2304 pixels ∗ 1296 pixels (W∗H)" - Is this the minimum image resolution for your model? Please specify the minimum image parameters required.
In lines 212-213 - Describe the subject of the research, specifically the abnormalities. As mentioned earlier, it would be beneficial to provide a more detailed description of the study's concept. What abnormalities are you aiming to identify based on feeding detection? Is it possible to deploy the algorithm for different types of animals or perhaps adult pigs? What changes would be necessary?
In line 216 - Can you justify what contributed to the improvement in mAP and F1 score parameters? Clarify the main differences between your model and the compared models. Does your proposed model have any weaknesses? If yes, please specify them.
In line 231 - Explain the term "missed detection."
In lines 231-237 - How did you address situations with pig's head overlap?
In line 233 - How did you ensure that false detections did not occur under head overlap?
Can you clarify the differences between the terms "overlap," "occlusion," and "occlusiveness"?
In line 238 - The abbreviation NMS is not explained.
Figure 12 - Texts near bounding boxes are not legible; try adjusting colors to enhance readability.
Include descriptions for parts a, b, c for Figure 12.
In lines 242-243 - It would be beneficial to provide clearer definitions for these functions (GIOU, DIOU, and CIOU) and describe their differences.
Table 4 - There is a typo in the table header. Instead of "Model," it is written as "Modle."
In line 247 - Based on the images, it appears to be enclosed spaces somewhere inside the hall. What impact does weather have on piglet detection during feeding?
In line 249 - Did the number of categories change compared to the previous article? In the previous article, there was a category for piglet heads in the feeder.
Figure 13 - Why aren't parts b and d described in the figure caption?
In line 250 - When using abbreviations FB and NFB for the first time, provide their full forms.
In line 251 - What is the purpose of this method and the overall approach? Please explain if it is solely about detecting piglet heads in low light conditions or if there is a specific reason for this approach. Providing a detailed description of the study's concept in the introduction of the article would address this issue.
In line 254 - "heatmap represents the regions of interest in the model" - Is this solution also used in cases with good lighting conditions? When do you apply this solution?
In line 260 - "the herd behavior of pigs and reflect the health status of pigs" - The article does not address this evaluation, and it would be beneficial to separate what you are counting (detecting piglet heads) from the potential use, such as behavior detection for health assessment. Theoretical insights on how this could be evaluated would be valuable.
In line 267 - "3D cameras have no advantage in the field of view" - The advantage of 3D cameras lies in spatial data (3D coordinates), which could be utilized in evaluating the relationship between piglet heads and the feeder.
In line 271 - "The total consumption time of food" - Please explain the significance of your study in this context. Specify whether the evaluation goes down to the level of individual piglets. If not, the evaluation could be limited to monitoring the cumulative weight of the feeder over time. Nowhere is it mentioned whether the results apply to specific individuals or to piglets as a whole.
In Figure 14 - Add units to the description of the axes, especially for the Y-axis. Explain what "total time of FB" means.
In Figure 15 - The Y-axis lacks a description.
Consider whether it would be more concise to represent time on the X-axis in hours from the beginning of monitoring (0, 1, ..., 24 [hours]) rather than the current format.
Author Response

(The authors gave the same response as above.)

Reviewer 4 Report
Comments and Suggestions for Authors
The study has merit but needs significant improvement to enhance the manuscript quality.
Decision: Major Revision
* Please use only abbreviation convention for example deep learning (DL) or Deep Learning (DL), please check for inconsistencies.
* The contributions are not detailed enough, please add more details and some background and justification of your contribution. Also present contributions in bullet form, not in a paragraph.
* The algorithm is not easy to understanding, it lacks important details, please add some information for ease of the reader.
* Your research closely aligns with the insights and methods discussed in several pivotal papers. To enhance the foundation of your study and provide readers with a broader context of Deep Learning Models, I recommend incorporating the following influential works in the introduction section:
1. https://doi.org/10.32604/csse.2023.037992
2. https://www.sciencedirect.com/science/article/pii/S0957417423009673
3. https://doi.org/10.32604/csse.2023.034475
* I also recommend to share the code publicly in the revised version for readers interest and research work integrity.
Author Response

(The authors gave the same response as above.)

Round 2
Reviewer 4 Report
Comments and Suggestions for Authors
The current manuscript has been revised carefully, therefore I recommend to publish article in its current form.
Author Response
.